# Consumer-Driven Demand-Side Management Using K-Mean Clustering and Integer Programming in Standalone Renewable Grid

**Muhammad Ahsan Ayub** [1], **Hufsa Khan** [2], **Jianchun Peng** [3,*] **and Yitao Liu** [3,*]

[1] College of Physics and Optoelectronics Engineering, Shenzhen University, Shenzhen 518000, China; ahsanayub@email.szu.edu.cn
[2] College of Computer Science and Software Engineering, Shenzhen University, Shenzhen 518000, China; hufsakhan@email.szu.edu.cn
[3] College of Mechatronics and Control Engineering, Shenzhen University, Shenzhen 518000, China
[*] Correspondence: jcpeng@szu.edu.cn (J.P.); liuyt@szu.edu.cn (Y.L.)

**Abstract:** Many countries have larger land areas and scattered communities. Therefore, to electrify them, small standalone power systems are the more preferred and cost-efficient solution as compared to utility grid extensions. The main objective of a standalone power system is to supply cleaner, cheaper, and uninterrupted electricity. However, for standalone power systems, demand-side management always remains a challenging task. In this paper, a load scheduling algorithm driven by K-mean clustering and linear integer programming to schedule consumers' appliances for the upcoming day is proposed. In addition, the basic power to run the necessary appliances is kept available in the system all the time. Furthermore, to assist the consumer in every situation, the battery storage system and the overall system size reduction are also taken into consideration. Consumer input is also used in scheduling the appliances. The proposed method is evaluated on the publicly available real-world dataset; the simulation results demonstrate that the proposed approach performs better, due to which the reliability and continuity of the system are increased.

**Keywords:** standalone hybrid renewable energy system; K-mean clustering; demand-side management; demand response

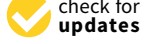



## 1. Introduction

The electrification of remote and islanded villages from the central grid system is a challenging and expensive task. It leaves 15% of the world's population without electricity [1]. Especially in remote areas, due to economic and geographic reasons, people face so many challenges related to energy supply [2]. In the current era, the usage of electricity in all countries is increasing day by day. In [3], it is mentioned that electricity consumption will increase by 53% in 2035. Therefore, the most challenging task for smart grids is to keep the balance between consumption and generation. This is particularly true for those residential areas which are located far away from the electricity network, with most of them using diesel generators [4,5]. There are some disadvantages in the use of these diesel generators, i.e., its negative impact on the environment, fuel carrying, and variation in fuel prices [6]. To resolve the preceding problems, the use of renewable energy sources for electricity generation has increased [7,8] since wind and solar are the two renewable energy sources that are ubiquitous in nature. These renewable energy sources are useful for the electrification of islanded areas and have many advantages, but at the same time, they also have some disadvantages such as their dependency on weather conditions and unpredictable nature. In addition, solar energy and wind turbines are unable to meet the consumption demand on critical days due to a lack of the required solar irradiations and wind speed [9]. The research community has devised different approaches to resolve

these problems; the most appropriate approach is demand-side management (DSM). We can overcome the power outage problem with the help of DSM and perform the optimal scheduling of appliances according to the weather conditions. The main objective of DSM is the continual availability of electricity for appliances of basic use, while other appliances are scheduled according to consumer convenience.

DSM has an important feature which is called demand response (DR) [10,11], and it deals with the residential, commercial, and industrial sectors. In DR strategies, two important points are customer convenience in reducing energy consumption as well as the load curve peak shaving and valley filling. In the existing literature work related to standalone systems, most of the work is focused on DSM and DR in relation to the economic point of view. In [12], appliance usage is defined based on time-varying electricity tariffs and coordination between grid supply, photovoltaic, and battery. An analysis of different techniques based on dynamic pricing tariffs is summarized in [13] for shaving peak hours and reducing load usage. In the literature, different optimization techniques for DSM have been proposed in which advanced metering infrastructure-based energy management [14], evolutionary algorithm [15], and binary partial swam optimization heuristic algorithm [16] are included. In all of these previously mentioned papers, the load shifting and control techniques are set forth without consideration of customer behavior towards insufficient electricity supply and generation constraints [17].

In [18], a novel approach based on the Dijkstra algorithm was proposed to reduce the cost for customers by adopting the less computational complexity strategy. In [19,20], the authors proposed an autonomous DSM system based on game-theoretic energy consumption scheduling for energy demand in order to reduce the overall energy costs and the peak-to-average ratio (PAR) of the total energy demand. Moreover, in [21], a genetic algorithm was used to minimize the energy consumption of commercial, industrial, and residential buildings. A modified mild intrusive genetic algorithm (MMIGA) was developed for off-grid residential buildings [4], and an ant colony algorithm was proposed to solve the optimization scheduling problem [22]. It optimizes the overall cost of the system by considering load demand, electricity wholesale market prices, meteorological data, and the operation of the micro grid for 24 h.

However, for the last several years, linear programming (LP) has remained an attractive field for the implementation of DSM [23]. In [24], an optimum energy management system for a small business or household using LP was proposed. In [25], a hybrid iterative load disaggregation algorithm using LP and clustering techniques was proposed to achieve better accuracy for all types of household appliances. Mixed-integer linear programming (MILP) was used in [26] to obtain the minimum household payment and to decrease the PAR of the load profile. In [27], an MILP formulation was presented to control the energy consumption price of the household appliances based on electricity tariffs. The objective function and constraints in MILP are linear equations, while the variables can be real or integer values [28]. In [29], an effective approach was presented to reduce the power and energy losses in a distribution network by using the storage systems. In [30], the positive effects of demand response and storage systems on the optimal management of a smart home were evaluated. In [31], a demand management system using convex programming for the energy storage system was proposed for a house with a battery and an electric vehicle. The authors of [32] provided a detailed analysis examining the effects of battery storage on micro grids to reduce their operational costs.

Another problem in optimizing the standalone grids is big data processing. To analyze the high-volume datasets [33], traditional methods for sample averaging are not a suitable choice [34,35]. However, in smart grid systems, analyzing the data based on machine learning algorithms has been gaining more attention. Clustering is one of these machine-learning algorithms which divides the dataset into different groups based on the similarity of a data point. Currently, for electrical power plant issues, the use of clustering has been increased. Therefore, many systems have analyzed load consumption data with the help of

clustering. For example, in [36], with the help of fuzzy c means and linkage clustering, the load data from seven regions in Italy were investigated to achieve significant output.

In [37], dynamic clustering was used to get the peak hours of consumption patterns. In [38], K-means and hierarchal clustering were used to cluster the substations into some categories according to their load profiles scanned shape. In [38], K-means and hierarchical clustering were used to compare their results for load profiles. In [39], with the help of clustering, commercial load profiles were analyzed. In [40], clustering was used for the coordination of wind turbines. In addition, the power generation and wind speed of solar-based distribution networks were analyzed by using Linkage-ward clustering [41]. In addition, clustering was used to analyze the commercial load profile [39] and also to analyze the behavior of renewable sources, just as in [40] where clustering was used for the coordination of wind turbines. In [41,42], wind power generation and quality were analyzed by using fuzzy c mean clustering [42]. In the existing research work related to DSM, most of them lack the modeling of hybrid energy resources [43]. The MILP base problem was proposed in [44] for off-grid residential houses which contain PV and batteries; by considering the supply and demand constraints, it was formulated that the load demand must be less than the available power all the time. However, the power supply from these constraints was continuous, and it did not define the best time for using controllable appliances. In [45], the linkage-ward clustering method was used to define time-varying tariffs and schedule the appliances according to the pre-determined hourly energy generation. Clusters were formed using the probability of their occurrence, and objective function was formulated using time-varying tariffs. Existing research works designed on the basis of time-varying tariffs lack consumer input as a variable function in scheduling the appliances. Furthermore, the time-varying tariffs increase the size of the generation system, and as a result, the overall capital cost of the system is increased.

In this work, a DSM algorithm is proposed as shown Figure 1. By using renewable resources such as PV and wind turbines for those houses that are located far away from the main city. A battery storage bank is included in the system to provide the baseload power at the time when renewable sources are not sufficient to meet the baseload power demand. Diesel generators equal to the rating of the baseload are kept as backup to avoid total black out in the worst conditions. Our objectives are to schedule and shift the controllable appliances by estimating hourly power generation and by considering consumer demand as a variable function. The appliances are prioritized based on their need and demand to achieve optimum energy utilization.

The main contributions of this paper include the following:

- A demand-side management algorithm is proposed to fulfil the energy gap between generation and consumer demand for standalone renewable energy systems;
- K-mean clustering is used to group the data based on two factors: probability of turning on a specific appliance at time $t$, and the priority number given by the consumer to that specific appliance;
- Linear integer programming is used to schedule the clusters of appliances based on the available power and state of charge of the battery system.

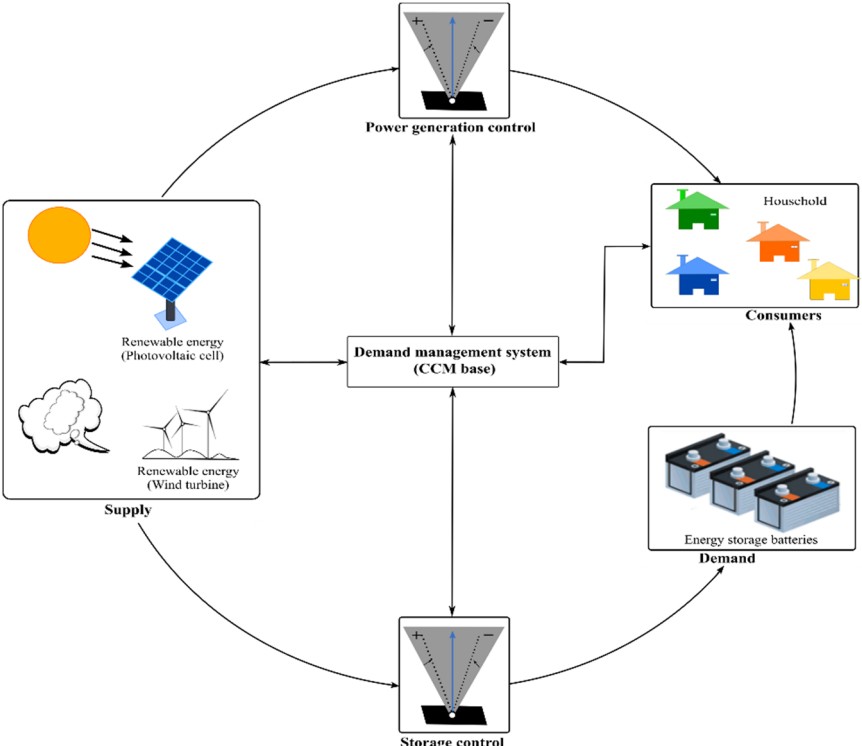

**Figure 1.** Block diagram of the proposed method.

The rest of the paper is organized as follows: Section 2 explains the modeling of electricity generation sources in the system and the factors affecting their output. The system model and problem formulation are presented in Section 3. Our proposed demand management system is presented in Section 4. The experimental setup, results, and empirical analyses of the proposed approach are reported in Section 5. Finally, the concluding remarks are given in Section 6.

## 2. Mathematical Modeling of the Energy Sources

Standalone renewable energy sources considered in this work are: solar panels, wind turbines, battery storage systems, and diesel generators. Parameters affecting the generating capacity of each source are discussed in the upcoming sections.

### 2.1. Solar Panels

Photovoltaic cells combined in panels are used to convert solar energy into electricity. Panels are stacked in series to form strings, and these strings are connected in parallel to build PV arrays. The number of PV arrays is increased to achieve the required level of electricity production. Solar irradiance falls onto the PV arrays, which converts solar energy into DC voltage and current. After the installation of the PV arrays, solar irradiance and temperature are considered as the key factors in the production of electricity. Based on different locations, losses of up to 10% are considered due to dirt and snow. Although sun-tracking features are added to increase the output of a single array, in this study, we used a fixed optimized installation position for PV arrays.

The output power of the PV arrays fluctuates due to weather conditions, solar irradiance, and temperature. Therefore, considering all PV strings are exposed to the same conditions, the approximate output power of PV arrays can be calculated by using Equation (1) [46]

$$P_{solar} = n \times P_{solar,rated} \times \frac{S(t)}{S_{ref}} \times \eta_{loss} \times \eta_{conv}^{PV} \tag{1}$$

where $P_{solar}$, $P_{solar,rated}$, and $n$ are the PV arrays' output power, the rated power of a single PV array, and the number of arrays in the system, respectively. Furthermore, $S(t)$ and $S_{ref}$ are the solar irradiance on the PV module surface (kW/m$^2$) at time $t$ and the reference solar irradiance (1.5 AM), respectively. In addition, $\eta_{loss}$ and $\eta_{conv}^{PV}$ provide the efficiency standing for the loss due to temperature and the DC–DC converter efficiency, respectively.

$$\eta_{loss}(t) = 1 - \lambda(T_{cell} - 25)$$
$$T_{cell}(t) = T_a(t) - \frac{S(t)}{0.8} \times (T_{NOCT} - 20) \tag{2}$$

where $\lambda$ stands for the temperature coefficient (0.00485/°C), $T_{cell}(t)$, $T_a(t)$ and $T_{NOCT}$ are the temperature of the PV cell, the ambient temperature, and the nominal operating temperature (i.e., 45 °C), respectively. By using Equations (1) and (2), the PV system is designed according to the consumer load whose annual load peak curve is the highest out of all the consumers in the community.

### 2.2. Wind Farm

Wind is another renewable source of electricity, available at most places. Wind turbines are used to extract energy from the wind and convert it into electricity. The rotor blades, gearbox, and generator are the main parts of the wind turbine. The power produced from wind is estimated by using Equation (3):

$$P_{wind} = \frac{1}{2}\rho A V^3 C_p \tag{3}$$

where $\rho$ is the air density (i.e., 1.255 kg/m$^3$), $A$ is the area swept by the rotor blades in m$^2$, and $V$ is the velocity of air in m/s. It can be seen that wind power ($P_{wind}$) is directly proportional to a cube of air velocity. In addition, the actual power extracted from the wind is the difference between upstream and downstream wind power. In Equation (3), $C_p$ is the fraction of upstream wind power that is converted into electricity by the blades, and the remaining power is wasted in the downstream wind. Here, $C_p$ represents rotor efficiency, and theoretically, the maximum value of rotor efficiency is 0.59. In particular, it varies from 0.2 to 0.5, depending on the turbine speed and the number of blades. In this study, we assumed it to be 0.45.

$$P_{wind}(t) = \begin{cases} 0 & ; v(t) \leq v_{in} \text{ or } v(t) \geq v_{out} \\ P_{wind,rated} \times \frac{v(t) - v_{in}}{v_r - v_{in}} & ; v_{in} \leq v(t) \leq v_r \\ P_{wind} & ; v_r \leq v(t) \leq v_{out} \end{cases} \tag{4}$$

where $v(t)$ is the wind speed at time $t$, $v_{in}$ is the wind turbine's minimum operating speed, $v_{out}$ is the wind turbine's maximum operating speed, and $v_r$ is the rated wind speed for the turbine. The capacity of the wind turbine installed is equal to the load of the consumer whose annual peak load curve is the highest in the community.

### 2.3. Battery Storage

A storage system is an essential part of the renewable standalone system that bridges the gap between the base power and the generated power when no renewable energy source is available for conversion. Batteries are the most preferred storage system as they are easy to install and scalable for every type of load, with a 6–10-year lifespan.

The efficiency of the battery system depends on two parameters: the first one is the state of charge (*SOC*), and the other one is the depth of discharge (*DOD*). These two parameters are controlled to increase the battery lifetime and system reliability.

Since it depends on the yearly weather forecast for the selected site, the battery storage system is designed to fulfil the baseload demand of the community throughout the year. In this work, two assumptions are made related to the dynamic behavior of battery storage:

1. The *SOC* of the storage battery changes linearly, and the charging and discharging power are uniform;
2. The charging and discharging efficiency of the battery is 100%.

$$P_{Batt} = (SOC - DOD) \times P_{rated(battery)} \tag{5}$$

### 2.4. Diesel Generator

A non-renewable energy source such as a fossil fuel-consuming generator is also used as a backup and for emergency situations. The benefit of a diesel generator is that it is detachable and easy to attach to the system when needed again. On the other hand, its drawback is that it consumes fossil fuel, which causes pollution and has worse effects on human health and the environment. Currently, it is widely used in remote areas to generate electricity and make the lives of the inhabitants easier. The electricity production cost depends on the efficiency of the generator and the fuel used. Although generators make human life easier, they are still expensive and not affordable for everyone.

In our proposed management algorithm, we minimize the use of diesel generators in the system. The proposed system maintains the minimum electric power required by every consumer in the system all the time. This is the power needed by basic household appliances such as tube lights, refrigerators, and fans. A diesel generator is included in the system to compensate for a total blackout situation. The capacity of a diesel generator is equal to the sum of the energy requirements of basic appliances of all houses in the community.

## 3. System Model and Problem Formulation

Generation systems are always designed to match the load from peak hours, which results in the wastage of energy in off-peak hours. A better energy demand management system reduces the wastage of energy in off-peak hours and decreases the capital investment of the system as well. This paper proposes a demand-side management system based on two algorithms: first, the K-mean clustering algorithm based on appliances' weights, and second is the linear integer programming (LIP) algorithm that schedules the appliances according to the available power in the system. Further details on these two algorithms are discussed in the following sections.

### 3.1. Probability Weights

Most of the consumers feel comfortable in using their appliances at a time of their choosing. As the restriction on the usage of appliances without considering consumer consent is not acceptable to the community, in this paper, consumer choice is considered in the scheduling of their appliances. Each appliance is given a weight based on its probability of usage at that hour, on a specific day and month. Available renewable energy is allocated among the consumers' appliances whose probability of usage at that specific hour is the highest based on previous data.

Let $A_i$ be the $i$th number of appliances at a consumer's house $c_j$, where $j$ is the house number. The probability of turning on the appliance $A_{i,j}$ at hour $h$ on weekday $Y$ is calculated by using the Bayes' theorem, as stated in Equation (6):

$$H = Event\ occurs\ at\ hour\ h$$
$$Y = Day\ of\ the\ week \tag{6}$$
$$w_{i,j}(H/Y) = \frac{\Pr(H,Y)}{\Pr(Y)}$$

### 3.2. K-Mean Clustering

The consumer data are categorized into a matrix $\Re^{j \times i}$ based on the probability weights and consumer priority. The number of appliances in a house is represented by $A = [a_1, a_2, \ldots, a_i]$, and the appliance probability weights in house $j$ are denoted by

$W_j = [w_1, w_2, \ldots w_i]$, where $i$ is the total number of appliances in a house. Consumers are represented by $C = [c_1, c_2, \ldots, c_j]$, where $j$ is the total number of houses in the community. The priority for each appliance as set by the consumer $j$ for an appliance $i$ is denoted by $E_{j,i} = z$ where $z = 0 \rightarrow 1$. For example, $E_{1,2}(t) = 0.6$ shows that consumer 1 set the priority at 0.6 for the second appliance at time $t$.

In particular, consumer appliance data are denoted by Equation (7):

$$D_j(t) = [d_{1,j}(t), d_{2,j}(t), \ldots, d_{n,j}(t)] \tag{7}$$

$$d_{i,j}(t) = w_i(t) + E_{i,j}(t) \tag{8}$$

where $j$ is the number of houses in the community, and $d_{i,j}$ is the array containing the probability weight of appliances $i$ and its priority number at time $t$. Time is divided into each hour of the day for 365 days; therefore, in total, we have 8760 timestamps for a year. The consumer can update their appliance preferences once a month for each day of the week.

In addition, the K-mean clustering algorithm is applied to the data $D = \{d_1, d_2, \ldots, d_n\}$ to make $k$ number of clusters on the basis of appliance data at time $t$ ($d(t)$). For a given set of data, K-mean clustering returns the k cluster centroids $V = \{v_1, v_{2,}, \ldots, v_k\}$. Through this clustering algorithm, we get the clusters for each hour and each day of the week as the number of clusters for each timestamp is greater than 1, and the maximum number of clusters that can be formed is equal to the total number of appliances in the community. The distance of the data point $d_{i,j}$ from the centroid of each cluster is calculated by using the Euclidean distance formula. The complete clustering procedure used in this study is illustrated in Algorithm 1.

---

**Algorithm 1:** K-mean clustering algorithm

---

**Input:** Appliance data of each consumer $D_{i,j}(t)$
Initialize: select $k$ number of clusters ($C_k$) with their centroid value $V_k$
Flag = 0

1.     **While** cluster size remains unchanged **do**
2.       Calculate the Euclidean distance of each data point from the mean of cluster $\|d_{i,j} - V_k\|$
3.       Assign the data point to the cluster whose Euclidean distance with mean is minimum
4.       Recalculate the cluster means
5.       Calculate the size of the clusters
6.       **If** the cluster size is unchanged **then**
      Set Flag = 1
      Terminate the **while** loop
      **end if**
7.       Update: cluster means
8.     **end while**
9.     Arrange clusters in an array **C** in descending order with respect to their centroids
10.    **Output:** Array containing $k$ number of clusters in descending order of their centroids
      $C(v) \, \forall \, v = 2, 3, \ldots, n$.

---

### 3.3. Battery Management System

The controlled charging and discharging of batteries will increase the reliability of the system and the lifetime of the batteries. Battery power is reserved for fulfilling the basic load requirements for 24 h, and the excess power is distributed among the consumers. However, batteries are charged on a priority basis if the supply of basic energy is not available for 6 h before scheduling the consumer appliances, in which case *DOD* is considered in the calculation to increase the efficiency of the batteries. The *SOC* of the battery is also a decision variable in determining the storage power. The reserved power in the battery $P_{reserved}(t)$ is calculated by using Equation (9). $P_{generated}(t)$ is the schedulable power available at time $t$

after subtracting the baseload power $P_{basic}$ from the total generated power. The power is reserved in the storage system to supply the base power for an uninterrupted 6 h period.

$$P_{reserved}(t) = 6 * P_{basic} - P_{Batt}(t) \tag{9}$$

$$P_{generated}(t) = P_{wind}(t) + P_{solar}(t) - P_{basic}(t) \tag{10}$$

$$Z(t) = P_{generated}(t) + \partial(t) * P_{battery}(t) \tag{11}$$

where $\partial$ is the sign with $P_{battery}$, which represents the battery condition either in its charging state or discharging state.

$$\partial(t) = \begin{cases} +ive & if\ P_{generated}(t)\ or\ P_{reserved}(t) < 0 \\ -ive & if\ P_{generated}(t)\ and\ P_{reserved}(t) > 0 \end{cases} \tag{12}$$

### 3.4. Linear Integer Programming

Total power available at time $t$ is distributed among the appliances using the linear integer programming (LIP) algorithm. The objective function of the LIP is to minimize the difference between available power and rated appliance power scheduled at time $t$. The constraints of LIP restrict the scheduling of more than two appliances for one consumer. This helps in distributing the available power to more consumers.

The total power available at time $t$ is represented by $Z(t)$, and $P(t)$ is the power calculated by LIP for the appliances that can be scheduled at time $t$. Furthermore, the $p$ array contains the power rating of the available appliances in the community, and $p(v)$ contains the appliance power rating in the cluster $v$. In addition, binary variables $X_{i,j}(t)$ are used to denote the state of the $i$-th appliance in $j$-th house at time $t$. The algorithm runs for a period of 24 h and schedules the appliances for each hour in advance.

$$X_{i.j}(t) = \begin{cases} 1 & if\ appliance\ \boldsymbol{i}\ in\ house\ \boldsymbol{j}\ is\ schudule\ in\ time\ \boldsymbol{t} \\ 0 & otherwise \end{cases} \tag{13}$$

$$P(t) = \sum_{j=1}^{n} P_j(t) \tag{14}$$

$$P_j(t) = \sum_{i=1}^{n} X_{i,j}(t)p \tag{15}$$

The objective function is formulated by using Equation (16), subject to constraints in Equations (17) and (18).

$$\underset{X_{i,j}(t)}{\text{min}imize} \sum_{t=1}^{24} (Z(t) - P(t)) \tag{16}$$

$$\sum_{i=0}^{n} X_{i,j}(t) \le 2 \tag{17}$$

$$X_{i,j}(t) \in \{0,1\}, \quad \forall \boldsymbol{i}, \boldsymbol{j}, \boldsymbol{t} \tag{18}$$

## 4. Proposed Demand Management System

In this section, we describe the proposed demand energy management system algorithm. The flowchart of the proposed system is shown in Figure 2. This algorithm is presented in two halves, as Algorithms 2 and 3. This system discloses the appliance scheduling to the consumer one day in advance. The weather forecast data for the next 24 h are obtained from [47] to predict the power generation $Z(t)$ for each hour. The predicted power is then scheduled for the consumers according to the clusters formed.

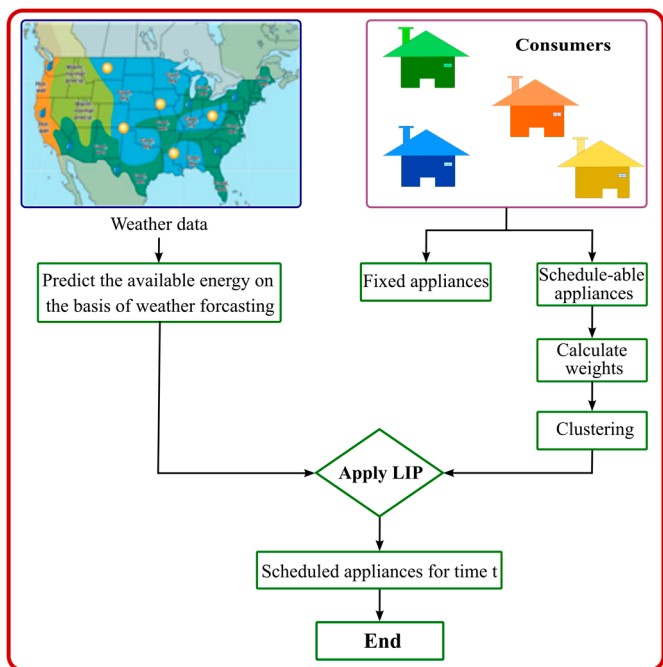

**Figure 2.** Work flow of the proposed management system.

Consumer appliances are divided into $v$ clusters based on their probability weights and consumer priority number for each hour. Since the centroids of each cluster vary between 0 and 2, where 2 is the maximum and zero represents the least important appliances at that hour. If the probability weight of an appliance is 1, and consumer priority is also set to 1, then it is placed in the cluster with a centroid approximately equal to 2. The likelihood of this appliance being turned on is the highest for that hour if power is available.

---

**Algorithm 2:** Categorization of Data

---

**Input:** Appliance consumption data of the community
       Consumer appliance preference for weekdays.

1.     Bayes' theorem is used to determine the weight of the appliances $w_i(t)$, $\forall t \in [H, Y]$
2.     Add appliance weight and consumer appliance to determine $d_{i,j}(t)$
3.     $k$ number Clusters are generated by using the K-mean clustering algorithm

**Output:** Consumer appliance data in the form of clusters.

---

Algorithm 3 runs to allocate the consumer's appliance usage timeline for 24 h. For each iteration $t$ of the algorithm, the solar and wind energy is predicted to calculate the total available power at that hour. At the start of this algorithm, $\overline{Z}(t) = Z(t)$ and $v = 1$, where $\overline{Z}(t)$ is the power fed to the LIP for the appliance schedule, and $v$ represents the cluster number. In particular, $\overline{p}(v)$ is the power rating of the appliances present in cluster $v$. Furthermore, the linear integer program runs in a loop to check the appliances in every cluster until $\overline{Z}(t) < \min p$, where $p$ is an array containing the power rating of all the appliances available in the community. However, in each iteration, the scheduled appliances are stored in an array $P(v)$, cluster number $v$ is increased by 1, and $\overline{Z}(t)$ is updated to the remaining power available for allocation after each iteration in hour $t$. If condition $\overline{Z}(t) < \min p$ fulfills and the for loop is terminated, the scheduled appliances are stored in an array $P(t)$ for time $t$, and the value of $t$ is increased by 1; the whole process goes on for the next hour until $t \leq 24$. Finally, the while loop terminates when $t$ reaches the number 24, and the output array $P(t)$ contains the appliance schedule for the next 24 h.

---

**Algorithm 3:** Demand Management System Algorithm

---

Initialize: $t = 1$, $Z(t) = Z(1)$, $\forall t = 1, 2 \ldots 24$
    Flag $= 0$

1. **while** $t \leq 24$ **do**
2. Predict the solar energy for time $t$ using Equation (1)
3. Predict the wind energy for time $t$ using Equation (4)
4. Use Equation (11) to determine the available power $Z(t)$ at time $t$.
5. Set $v = 1 \,\& \, \overline{Z}(t) == Z(t)$
6. **for** $\overline{Z}(t) < \min p$
7. Use LIP to determine $\qquad P_v(t) = \sum\limits_{j=1}^{m} P_j(t) \qquad$ where

   $P_j(t) = \sum\limits_{i=1}^{n} X_{i,j}(t)\overline{p}(v)$, $\forall j = \{1, 2, \ldots, m\}$
8. Update $\overline{Z}(t) = Z(t) - P_v(t)$
9. Update $v = v + 1$
10. Update $P(v) = [P_v(t)]$
    $\qquad\qquad_{at\ time\ t}$
11. **end for**
12. Update $P(t) = P(v)$
13. Update $t = t + 1$
14. **end while**
15. **Output:** Appliances schedule ***P(t)***.

---

## 5. Results

In this section, we present the simulation results to assess the performance of our proposed algorithm. For analysis, we considered a community with five houses, and each house has some schedulable appliances, as shown in Table 1. The load pattern of the five houses is taken from two publicly available datasets, i.e., UK-dale [48] and Ampds [49]. In this study, only those houses that have a minimum of six months of readings are considered.

**Table 1.** Schedulable appliances in each house.

| House | Boiler | DW | Micro | WM | HD | Oven |
|:-----:|:------:|:--:|:-----:|:--:|:--:|:----:|
| 1 | ✓ | ✓ | ✓ | ✓ | | |
| 2 | | ✓ | | ✓ | | |
| 3 | | | ✓ | ✓ | | |
| 4 | | ✓ | ✓ | ✓ | ✓ | ✓ |
| 5 | | ✓ | | ✓ | | ✓ |

We also considered a baseload power for every house that could not be scheduled and had to be available at all times. The appliances for each house such as lights, fans, charging plugs, and a refrigerator were considered as baseload, and the sum of their power ratings were termed as base power for each house. In this system, we assumed 1000 watts as base power for every house, and an uninterrupted supply of base power was assured for every house.

Although the main objective of our algorithm was to produce a continuous power supply for the baseload, the consumer's usage behavior for schedulable appliances was given priority in the proposed load management system. In this system, consumer preferences are taken for each day of the week once a month, and to further enhance the user experience, it can also be taken once a week. Consumer preferences are termed as consumer weight for each appliance, and its value varies between 0 and 1, where 1 is the maximum and 0 is the minimum.

The weightage of the appliance was calculated for each hour and each day of the week based on its usage history for the last six months using Equation (19).

We ran the proposed management system to schedule the data for one week in January to verify the results. The consumer-given appliance weights for the week under consideration are shown in Figure 3. To further compare the schedulable appliances with the consumer's priority, we analyzed the results for one day of the week, that is, Monday. For instance, if a consumer is required to use a washing machine only on Mondays, in that scenario, our proposed management system will make sure that the consumer's washing machine is scheduled for Monday according to the available power.

$$w_i = \frac{\text{Number of time appliance turn ON at specific time}}{\text{Total number of instances under consideration}} \tag{19}$$

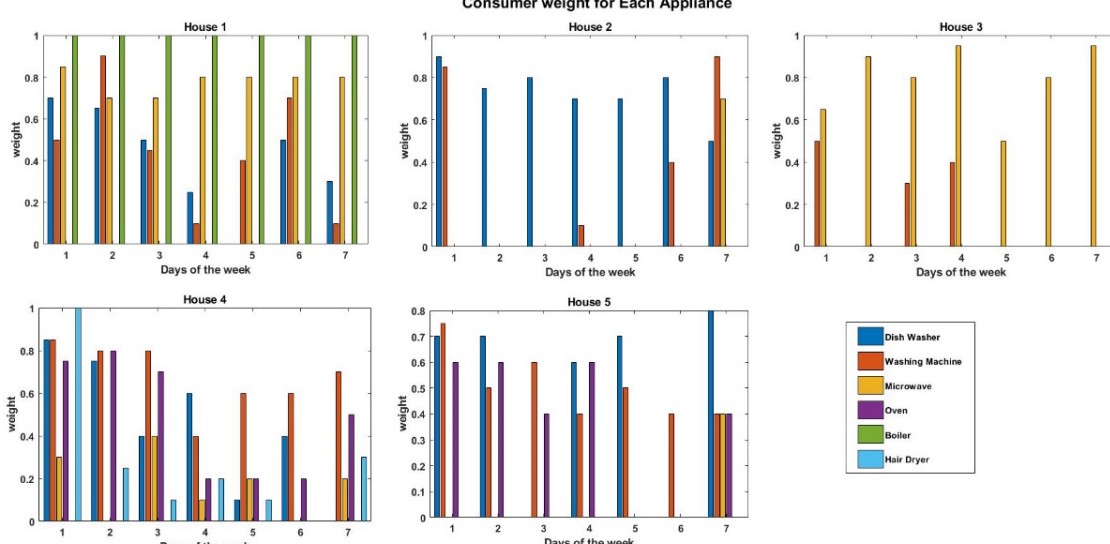

**Figure 3.** Appliances weightage given by each consumer.

The power rating of schedulable appliances in different houses under study are shown in Table 2.

**Table 2.** Power rating of the appliances available in the dataset.

| Appliance | Power Rating (watts) |
|---|---|
| Dish Washer (DW) | 2500 |
| Washing Machine (WM) | 2000 |
| Microwave (micro) | 1600 |
| Oven | 2200 |
| Boiler | 120 |
| Hair Dryer (HD) | 1100 |

In order to validate the proposed system, the simulation was performed on the MAT-LAB software, and the load profile of the houses were imported from publicly available datasets [48,49]. Renewable power generation was calculated using [46], and each renewable source was found to be capable of generating 50% of the peak power when running at its full capacity. The solar power generation in the month of January, calculated using the Ampds dataset, is shown in Figure 4; the average wind speed in the month of January is 13.5, as shown in Figure 5. The peak power generated by a renewable energy source in one week of January is 9500 watts, while the peak load demand for the same week is 16,000 watts, as plotted in Figure 6.

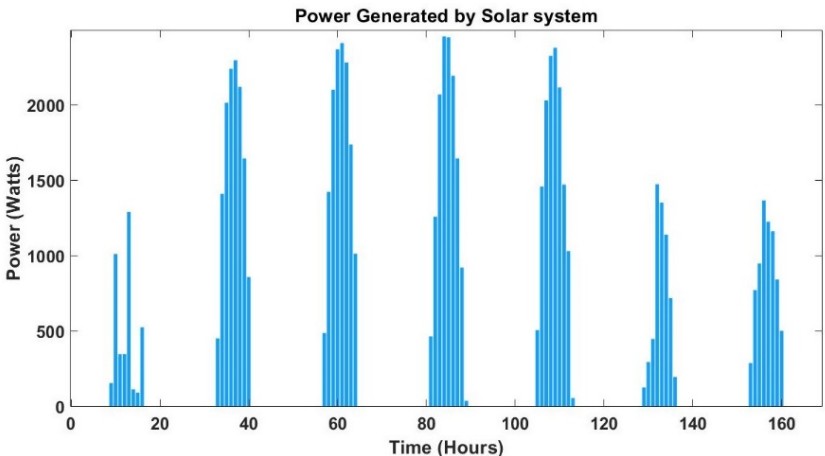

**Figure 4.** Power generated by the solar system.

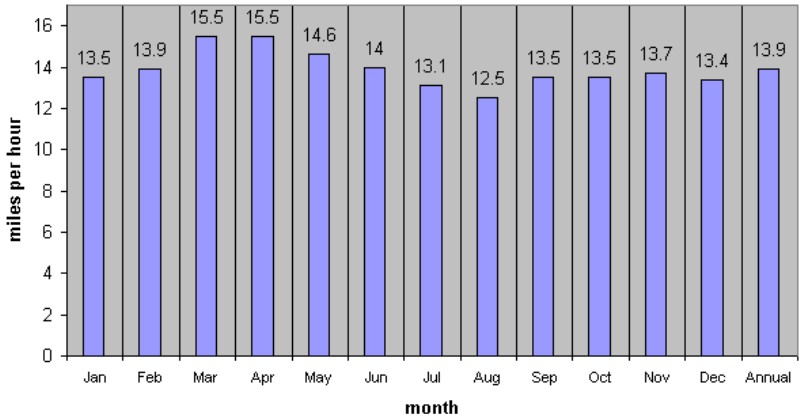

**Figure 5.** Monthly average wind speed data [50].

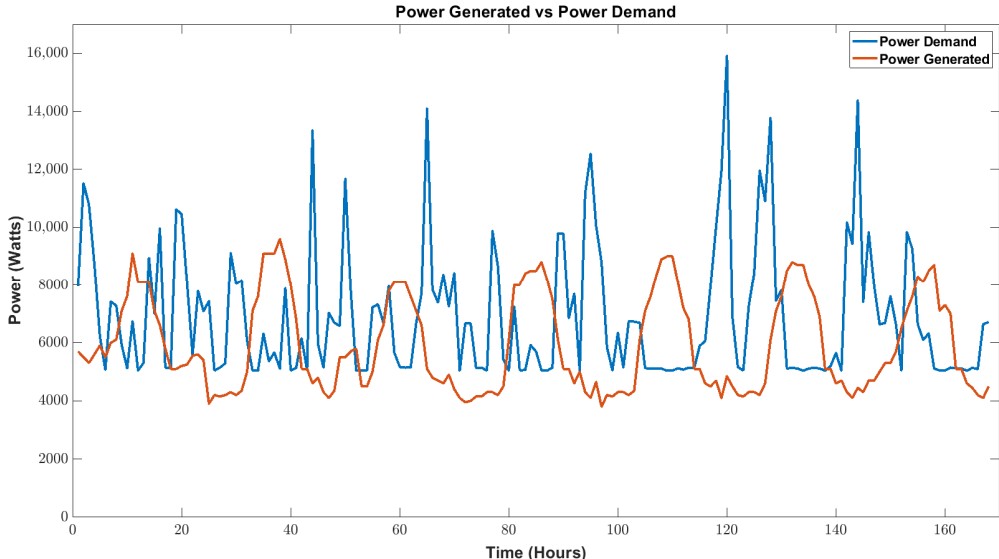

**Figure 6.** Power generated and power demand curves without the load management algorithm.

To fulfill the peak load demand, the off-grid renewable energy system increases its capital cost and wastes a lot of energy in the off-peak hours.

Our proposed load management system makes the curves for the power generated and the power distributed to the community align with each other, as shown in Figure 7.

In addition, the proposed approach helps in shaving the peak load curve and minimizing the wastage of energy during the off-peak period. Figure 7 shows the reduction in peak load and the new peak load demand at 9500 watts, which indicates a 40% overall reduction in the generation capacity. Figure 8 shows the charging and discharging patterns of battery banks, which are used to fulfill the baseload demand when less power is generated in the system and the excess power is used to charge the battery banks during an off-peak load period.

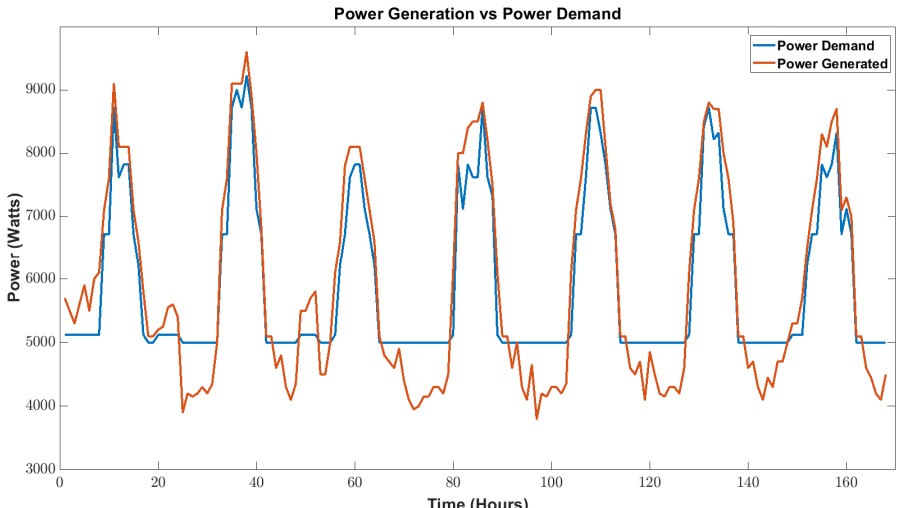

**Figure 7.** Power generated and power demand curves with the proposed load management system.

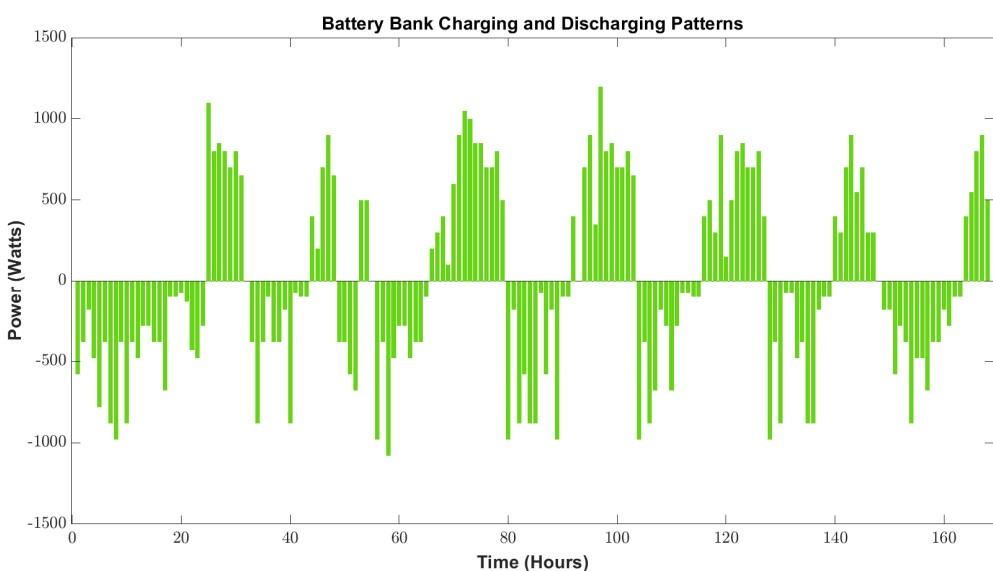

**Figure 8.** Battery charging and discharging states.

Furthermore, the battery banks are only used to fulfill the baseload requirements, which will decrease the storage bank capital cost and hence reduce the overall capital cost of the system by more than 40%.

In addition, to further validate the proposed system, the appliance schedule pattern for Mondays is drawn in Figure 9, and Figure 3 represents the appliance weightage given by consumers for Mondays. In addition, Figure 10 represents the power generated graph for the same day. The baseload is calculated as 1000 watts for each house to keep basic appliances turned on uninterruptedly. As there are five houses in the community, the total base power for the system is 5000 watts. The excess power is utilized to turn on the schedulable appliances on Mondays.

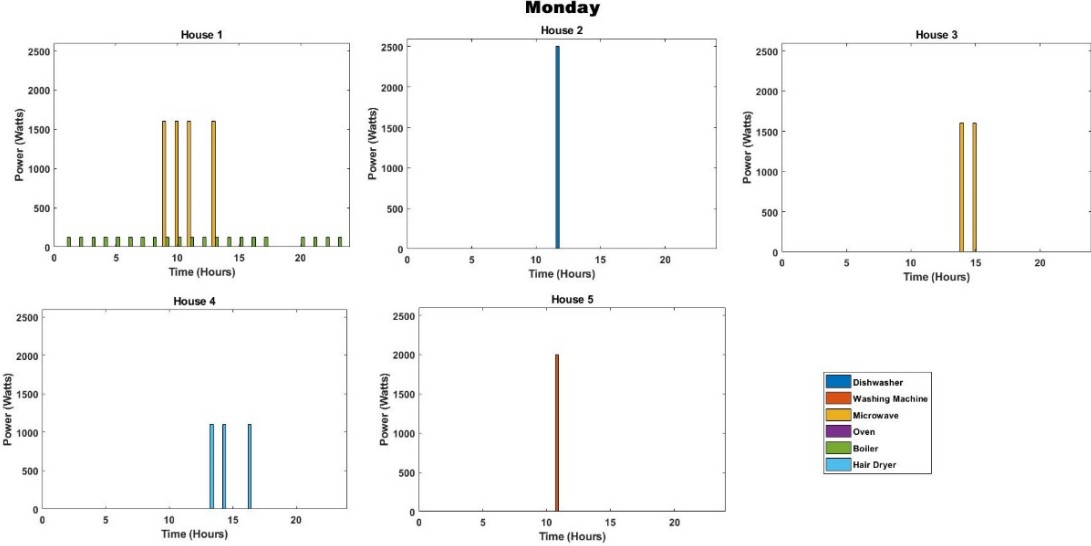

**Figure 9.** Appliance scheduled for Mondays.

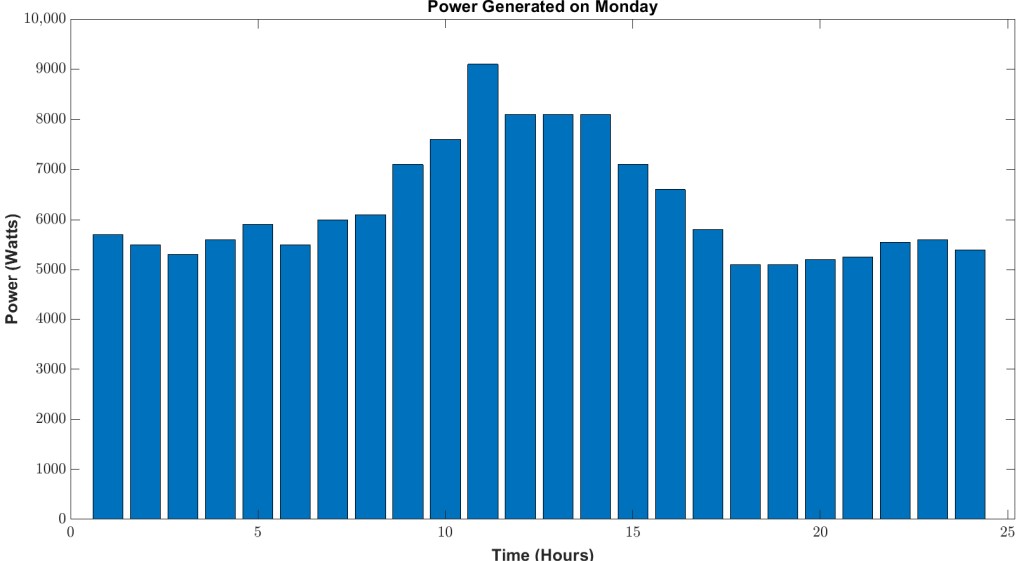

**Figure 10.** Power generated on Mondays.

If we compare the consumer appliance weightage in Figure 3 and the scheduled appliances for Mondays in Figure 9, the appliances that are given the highest priority by the consumer for Mondays are scheduled first whenever there is excess power available in the system. The appliance weightage not only determines the appliance priority in a single house but also which consumer appliances are scheduled first when there is excess power available in the system (as can be seen in Equation (18)).

In the previous literature, the load management systems were designed without considering the consent of the consumers. Therefore, adapting these systems is difficult for the consumer due to their work routines. In this situation, forced implementation methods such as increased tariffs for peak hours, forced load shedding, or forced power scheduling are adapted. In contrast, our proposed system will provide uninterrupted base power for each consumer in the community without any hike in power tariff and will schedule the appliances according to their usage habits. This will make the system practically more acceptable for the communities than the previously proposed load management systems in the literature. In addition, in the proposed system, the peak power curve is shaved, which decreases the capital cost of the whole system.

## 6. Conclusions

In this paper, we proposed a demand-side management system for a residential community electrified through an off-grid renewable energy system. The proposed algorithm assigned the available power to the appliances based on consumer usage data and consumer appliance priority using linear integer programming. Consumer preferences were taken once a month to increase the appliance usage convenience for the consumers. Furthermore, simulation results were generated for a community of five houses, which showed the acceptability and the possibilities for the practical implementation of our proposed management system in facilitating better energy utilization; it also enhanced the consumer's level of comfort. In future work, the consumer appliance weightage can be taken with respect to other houses in the community, and the frequency of consumer appliance weightage can be changed to once a week in order to increase the convenience for the consumers.

**Author Contributions:** Conceptualization, M.A.A.; methodology, M.A.A.; software, M.A.A.; validation, M.A.A., J.P. and Y.L.; writing—original draft preparation, M.A.A.; writing—review and editing, H.K. and J.P.; supervision, J.P. and Y.L.; project administration, J.P. and Y.L.; funding acquisition, J.P. and Y.L. All authors have read and agreed to the published version of the manuscript.

**Funding:** This research was funded by Foundation of Shenzhen Science and Technology committee and Foundation of Shenzhen Science and Technology under grant number GJHZ20180928160212241 and JCYJ20190808165201648 respectively.

**Acknowledgments:** This work was supported in part by the Foundation of Shenzhen Science and Technology Committee under Grant GJHZ20180928160212241, and in part by the Foundation of Shenzhen Science and Technology Committee under Grant JCYJ20190808165201648.

**Conflicts of Interest:** The authors declare no conflict of interest.

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
