# Peer review of "Consumer-Driven Demand-Side Management Using K-Mean Clustering and Integer Programming in Standalone Renewable Grid"

_energies, doi:10.3390/en15031006_

Round 1

Reviewer 1 Report

In this paper, a demand-side management system for a renewable-based isolated residential community is proposed. The proposed load scheduling algorithm is developed with the help of Kmean clustering and linear integer programming. Simulation results are presented to show the effectiveness of the proposed method. There are certain concerns.

  1. More recent related work should be referred. There is no reference from 2021.
  2. In the proposed system solar PV, wind generator and battery are the main generating system. As the proposed system is in isolated mode, how continuity of power supply be maintained? s
  3. The quality of the figures needs to be improved.
  4. Equations 4, 9, and 12 include some corrections and need to be modified.
  5. It could be better to include base load ratings in the results section (Fig: 10)
  6. Sizing of renewable energy sources and battery systems were not properly explored.
  7. Please discuss how consumers’ comfortability is marinated in the proposed method.

Reviewer 2 Report

- Authors should avoid using the Subjective pronoun “ we” in academic works.
- In such a study an experimental case must be presented in a specified region. 
-The authors should carefully distinguish the new contributions of their work from the new existing studies.
for example: 
DOI: 10.1016/j.est.2020.101221
 A comprehensive deeper literature review is necessary to address the research issue. Also, authors need to provide a literature survey in an organized way.
- All the figures need to be improved.

Round 2

Reviewer 1 Report

The authors have addressed the concerns. I do not have any more queries. 

Reviewer 2 Report

The manuscript has been revised. Unless the quality of figures is not improved.